# The Impact of COVID-19 on Health Behavior, Stress, Financial and Food Security among Middle to High Income Canadian Families with Young Children

**DOI:** 10.3390/nu12082352

**Published:** 2020-08-07

**Authors:** Nicholas Carroll, Adam Sadowski, Amar Laila, Valerie Hruska, Madeline Nixon, David W.L. Ma, Jess Haines

**Affiliations:** 1Department of Family Relations and Applied Nutrition, University of Guelph, Guelph, ON N1G 2W1, Canada; asadowsk@uoguelph.ca (A.S.); alaila@uoguelph.ca (A.L.); jhaines@uoguelph.ca (J.H.); 2Department of Human Health and Nutritional Sciences, University of Guelph, Guelph, ON N1G 2W1, Canada; vhruska@uoguelph.ca (V.H.); maddy.nixon@uoguelph.ca (M.N.); davidma@uoguelph.ca (D.W.L.M.); 3University of Guelph, Guelph, ON N1G 2W1, Canada; guelphfamilyhealthstudy@gmail.com

**Keywords:** COVID-19, family, health behavior, stress, food insecurity

## Abstract

The COVID-19 pandemic has disrupted many aspects of daily life. The purpose of this study was to identify how health behaviors, level of stress, financial and food security have been impacted by the pandemic among Canadian families with young children. Parents (mothers, *n* = 235 and fathers, *n* = 126) from 254 families participating in an ongoing study completed an online survey that included close and open-ended questions. Descriptive statistics were used to summarize the quantitative data and qualitative responses were analyzed using thematic analysis. More than half of our sample reported that their eating and meal routines have changed since COVID-19; most commonly reported changes were eating more snack foods and spending more time cooking. Screen time increased among 74% of mothers, 61% of fathers, and 87% of children and physical activity decreased among 59% of mothers, 52% of fathers, and 52% of children. Key factors influencing family stress include balancing work with childcare/homeschooling and financial instability. While some unhealthful behaviors appeared to have been exacerbated, other more healthful behaviors also emerged since COVID-19. Research is needed to determine the longer-term impact of the pandemic on behaviors and to identify effective strategies to support families in the post-COVID-19 context.

## 1. Introduction

The COVID-19 pandemic has disrupted the economic stability, stress levels, and daily routines of many Canadian families. While governments work to open our economies, infectious disease and economic experts have indicated that our lives will not simply return to our pre-COVID-19 normality. Our way of life has fundamentally shifted. In order to identify how best to support families in this post-COVID-19 context, we need to understand how these fundamental changes have impacted families in Canada.

The government-mandated physical distancing restrictions to reduce the spread of COVID-19 likely have had a considerable impact on families’ health-related behaviors. Limited access to outdoor recreational facilities, such as parks and playgrounds, reduces the opportunity for families to be physically active. Closures of schools and child-care centers may have further reduced children’s activity levels as these settings are shown to be associated with increased physical activity [1]. Reduced physical activity may also impact sleep quality and related routines [2]. Furthermore, Canadian food purchasing has shifted in response to COVID-19; revenues from dry goods, shelf-stable foods, and frozen produce have drastically increased relative to the average of the previous year [3]. This shift may influence family eating patterns and dietary intake. Lastly, as families navigate working and learning from home, time spent on screen-based devices has also reportedly increased [4]. Collectively, these changes may have a lasting impact on health outcomes among parents and their children. While existing research has explored changes in health behaviors among adolescents and youth due to COVID-19 [5,6,7], limited research has examined the impact of COVID-19 among families with young children. Given that the level of parental engagement in health behaviors is typically higher among families with young children versus those with older children and adolescents, the impact of COVID-19 may differ among these families as well. A clear understanding of how these changes have influenced the health behaviors, i.e., eating patterns, physical activity, sleep, and screen time, among families with young children is needed to inform family-based health promotion interventions that are relevant to the post-COVID-19 context.

In addition to the impact on health behaviors, COVID-19 also presents unique stressors that may impact families, including isolation or illness due to the virus, loss of employment with paralleled financial burdens, and coping with the abrupt shift in our everyday lives. A national survey among Canadians observed a significant decrease in mental health indices when compared to pre-COVID-19 benchmarks with 80% of respondents reporting the pandemic negatively impacting their mental health [8]. Knowing the impact that family-level stress can have on the health behaviors and outcomes of parents [9,10] and their children [11,12], it is important to understand how COVID-19 has impacted stress and financial concerns among families with young children. This information will guide efforts to support families in managing stress while simultaneously promoting healthy behaviors during this unprecedented time.

The objective of this study is to examine how health behaviors (i.e., physical activity, eating patterns, sleep, screen time) and level of family stress, financial and food security among a sample of Canadian families have changed since the COVID-19 physical restrictions have been implemented. Identifying the impact that COVID-19 has had on families with young children will help inform the development of effective, family-based health promotion interventions that are relevant in this post-COVID-19 context.

## 2. Materials and Methods

### 2.1. Study Design

The Guelph Family Health Study is a longitudinal family-based cohort designed to identify early life risk factors for chronic disease and to implement family-based health promotion strategies [13]. Families were eligible to participate if they had at least one child between 18 months and 5 years of age at the time of registration for the study, lived within the Guelph-Wellington area in Ontario and parents had to be comfortable with English to respond to survey questionnaires. Parents provided written consent in their initial study visit. The Guelph Family Health Study had two pilot phases (phases 1 and 2), which began in 2014 and 2015, respectively, in advance of the launch of the full-scale study, which began in 2017. This study used data collected from an online survey administered between 20 April 2020 to 15 May 2020 among parent participants in the Guelph Family Health Study pilot phase 1 and 2 and full study. The survey was sent to 306 families, which included 552 parent participants (310 mothers and 242 fathers) and 406 children. Completed questionnaires were received from 254 families (response rate = 83%), which included data on 235 mothers, 126 fathers, and 310 children. The first parent to enroll in the study was defined as Parent 1. Only Parent 1 was prompted to answer questions about the participating child(ren). Among the families where only Parent 2 responded to the survey (*n* = 19), child data were not collected. Parents received a $20 grocery gift card for completing the survey. The study was approved by the University of Guelph Research Ethics Board ([REB14AP008 and REB14AP009]).

### 2.2. Measures

The purpose of the survey was to understand the impact of COVID-19 on the health behaviors, specifically, eating patterns, physical activity, screen time, and sleep among parents and children, as well as stress, financial and food insecurity of families. Family sociodemographic data, i.e., age, ethnicity, household income, and marital status, were retrieved from previously administered surveys. 

#### 2.2.1. Change in Health Behaviors

For diet, parents were asked whether their eating has changed since COVID-19 (yes/no) with a similar question to assess whether their children’s eating had changed. If parents responded yes, parents were asked in what ways has their diet (or their children’s diet) changed (check all that apply): eating more/less food, eating more/fewer fruit and vegetables, eating more/less snack foods, such as chips or cookies, eating more/fewer foods from fast food/take out restaurants. We also asked about changes to family meal routines since the COVID-19 outbreak (yes/no). If parents responded yes, parents were asked in what ways have their meal routines changed (check all that apply): eating more/fewer meals with my child(ren), spending more/less time cooking, making more/fewer meals from scratch (meaning at least two ingredients combined), involving my child(ren) in preparing meals more/less often. For physical activity, sleep, and screen time, parents were asked whether time spent doing these behaviors had increased, decreased, or stayed the same since the COVID-19 physical distancing restrictions were implemented. Separate questions were asked for each of the behaviors and parents were asked these questions about their own behavior and that of their children.

#### 2.2.2. Stress

Using the scale from Parks et al. [14], parent-perceived stress over the past month was measured on a 10-point scale with ‘1’ indicating “no stress” and ‘10’ indicating “an extreme amount of stress”. Responses were reported with the mean and standard deviation of our sample. Parents also reported on their children’s stress with the following item: “How much has [child’s name] been troubled or worried about COVID-19?”. Response options included very little, somewhat, very much or I don’t know. 

#### 2.2.3. Financial Stress

Financial stress was assessed using two items adapted from the measures by Boushey and Gundersen [15]: (a) “During the past month, was there a time when you were worried you would not be able to pay the mortgage, rent or other bills on time?” and (b) “Are you worried about not being able to pay the mortgage, rent or other bills on time over the next 6 months?”. Response items included yes, no, or I don’t know. 

#### 2.2.4. Food Insecurity

Food insecurity was assessed using two items adapted from the measure by Gundersen et al. [16], these measures are as follows: (a) “During the past month, was there a time when you were worried you would not have enough money to buy food for you and your family?” and (b) “Are you worried about not having enough money to buy food for you or your family over the next 6 months?”. Response options included yes, no, or I don’t know. 

#### 2.2.5. Current Level of Health Behaviors among Parents and Children

Parent and child diet measures were adapted from the NHANES Dietary Screener Questionnaire [17]. Frequency, assessed by times per day, of fruit and vegetable and snack food intake (such as chips and cookies) were obtained. Fast food/take-out frequency was also assessed and was reported as average times per week. The International Physical Activity Questionnaire [18] was used to assess moderate-to-vigorous physical activity (MVPA), time spent walking and time spent sitting among parents. Child activity levels were assessed in terms of time spent in active play [19] and time spent playing outdoors [20]. Parents were asked to report total recreational screen time on average over the past month, for themselves and participating child(ren) [21]. Sleep duration for both parent and child participants was measured using items from the Pittsburgh Sleep Quality Index [22].

#### 2.2.6. Qualitative Questions on COVID-19

Parents were asked three open-ended questions: (1) “Please share anything else you would like to about how your child(ren)’s behaviors, i.e., their eating, physical activity, screen time, and sleep, has changed due to COVID-19”, (2) “Please share anything else you would like to about how the COVID-19 pandemic has impacted the stress you and your family are feeling or dealing with”, and (3) “What resources would help your family to deal with changes your family is experiencing due to COVID-19”. 

### 2.3. Data Analysis

Quantitative data from the study questionnaire and previously administered surveys were compiled and measured using descriptive statistics through R (version 3.6.1, R Core Team, Vienna, Austria, 2019). Qualitative data were analyzed using Nvivo (version 12, QSR international, 2020). Thematic analysis, as described by Braun and Clarke [23], was used to analyze responses to the open-ended questions in the survey. The qualitative analyst (AL) read all responses before beginning to code all comments. Coded data were grouped into common themes; initial themes were reviewed and edited (for example, renamed or collapsed) until each theme described a unique aspect of the data.

## 3. Results

### 3.1. Family Characteristics and Current Health Behaviors

The average age for children was 6 years with a standard deviation (SD) 2.0 years (Table 1). Mothers’ mean age was 37 (4.8) years, and fathers’ mean age was 39 (5.5) years. The majority of our sample identified as Caucasian (mothers, 87.0%; fathers, 88.0%) and over half of the families reporting an annual income of $100,000 or more.

Mean daily fruit and vegetable intake was 4.1 servings for mothers, 3.8 for fathers, and 4.5 for children (Table 1). Mean daily snack food intake was approximately one serving per day for mothers, fathers, and children. On average, mothers and fathers reported approximately 4.5 h and 9.5 h of moderate to vigorous activity per week, respectively (Table 1). Mothers reported sitting for approximately 6 h per day and fathers reported sitting for 6.5 h per day. Children’s time spent outdoors and time in active play were both reported as approximately one hour per day. Recreational screen-time was observed to be highest among fathers with a mean of 2.8 h per day, followed by mothers with 2.7 h per day, and children having approximately 2 h per day. The average sleep duration was for mothers roughly 8 h, for fathers just under 8 h, and for children almost 11 h/night.

### 3.2. Change in Health Behaviors Since COVID-19

More than half of our sample (mothers, 70%; fathers, 60%; children, 51%) stated their eating has changed since COVID-19 (Figure 1); the most commonly reported eating behavior changes were eating more food (mothers, 57%; fathers, 46%; children, 42%), eating more snack foods (mothers, 67%; fathers, 59%; children, 55%) and eating fewer foods from fast food and/or take out among parents (mothers, 43%; fathers, 45%). Nearly 60% of mothers and 50% of fathers reported that their meal routines had changed since COVID-19; the most commonly reported changes to their meal routines included spending more time cooking (mothers, 70%; fathers, 68%), making more meals from scratch (mothers, 65%; fathers, 58%), eating more meals with children (mothers, 60%; fathers, 53%), and involving children in meal preparation more often (mothers, 53%; fathers, 47%) (Figure 2). Since COVID-19, screen time has increased among 74% of mothers, 61% of fathers, and 87% of children and physical activity had decreased among 59% of mothers, 52% of fathers, and 52% of children (Figure 3). Many families reported that sleep duration had stayed about the same since COVID-19 (mothers, 42%; fathers, 49%; children, 68%), though some parents reported decreased sleep duration (mothers, 36%; fathers, 30%).

### 3.3. Family Stress, Financial and Food Security

Parents reported moderately high levels of stress (mothers: mean 6.8; fathers: mean 6.0; Table 2). Parent perception of child’s concern of COVID-19 indicated roughly 49% showing very little concern, 38% being somewhat concerned, and only 7% as very much concerned. Approximately 19% of mothers and 14% of fathers reported experiencing financial stress in the past month. Reported financial stress increased when regarding the next 6 months among both mothers (22%) and fathers (18%). Approximately 10% of mothers and 5% of fathers reported food security concerns either in the past month or over the next 6 months.

### 3.4. Qualitative Questions on COVID-19

Of the 361 parents who completed the survey, 182 responded to the open-ended questions. Responses to the open-ended question about changes in child behavior resulted in themes of change in children’s mood and general behavior as well as their health behaviors, in particular changes in physical activity and screen time. Responses to the question about family-level stress included themes related to factors that increase family stress, as well as themes related to strategies parents are using to cope with stress. In response to the question about supporting families, parents provided ideas for resources that would be helpful to them during the COVID-19 pandemic. This section will only focus on the themes that were most commonly reported.

#### 3.4.1. Physical Activity and Screen Time Changes

Many parents expressed concern about the changes to their children’s physical activity and screen time. Parents reported that physical activity is limited due to the lack of space and variety in available tools and toys: “*They also have no organized sports. For [daughter’s name], she used to do 6 hours of gymnastics a week - this is hard to replicate at home*” and “*They can really only play in the backyard, go for a walk or a bike ride*”. Parents also mentioned the intensity at which children are doing physical activity is less than before the pandemic: “*We go on daily walks but she isn’t running around like she might at recess*”. 

Parents reported an increase in screen time due to online learning: “*Screen time with school has increased tremendously. She spends about 3 hours on a device a day doing schoolwork”, playing video games: “playing games with her cousins on Kids messenger and drawing pictures on my phone for her cousins—this is also new”, watching TV: “He has wanted to watch more TV”, communicating with friends and family: “She’s used a lot of Messenger for kids to connect with family and friends” and to maintain their usual activities: “She didn’t want to do any extracurricular activities likes dance or karate for the first 3 weeks but then she got used to her instructors online and will now participate”. Many parents also reported that screen time has increased due to parents’ need to work from home: “They have had a tone of screen time. They had nothing before and now with both parents working from home..they have a lot!*”.

#### 3.4.2. Children’s Mood and General Behavior

Parents identified a common concern that their children were misbehaving more since COVID-19: “*Irritability has increased. Her patience to handle small troubles has diminished and she is more easily frustrated when things don’t go how she wants them to”. Some parents attributed these behaviors to being out of boredom: “She has been testing behavioral boundaries, seemingly out of boredom” and “They are restless, not dealing with boredom well (eating, fighting, bugging each other or us)”. Children were also saddened from not being able to see their friends and family: “They both miss their friends and find that aspect the hardest” and “She is deeply missing her family and friends*”.

Other parents reported that their children were doing well: “*She is acting very normally, and has decreased screen time in general. She is happy, sleeping and eating well, and not minding the change to routine at all”. Some children were enjoying staying at home: “He is rather enjoying staying and playing at home with us”, whereas others were appreciating the slower pace of daily life: “I can see that he has enjoyed a slower morning pace in the mornings than the rush hour we used to have to get ready for work and school*”.

#### 3.4.3. Factors that Increase Family Stress

There were several factors parents reported as increasing their stress. Many parents discussed finding homeschooling difficult, especially in addition to working full-time hours from home. Parents are also finding it difficult to balance their parenting role (including housework, keeping children engaged in activities, and homeschooling) alongside their work duties: 

“*Managing the needs of a preschool-aged child while also balancing the needs of 2 kids who are completing online learning assignments has been very challenging*”.

Some parents also reported working longer hours: “*For my teleworking and the project I work on resulted in more stress and longer work hours*”. Parents who are essential frontline workers reported that they are concerned about contracting the virus: “*I worry that I may take the virus home to my kids and their dad who lives at another home and works in long term care*”. In addition, partners of essential frontline workers also expressed worries of catching the virus:

“*My husband is a physician who works in a family medicine clinic, ER, and with inpatients in the hospital… We’ve had to navigate how to keep him as protected as possible, especially when he comes home after an ER shift to our family*”. 

In general, parents expressed concern of contracting the virus by being outside the home: “*Feeling of bringing home the virus from the grocery and sanitizing our grocery items*”; the unpredictable nature of the symptoms: “*The stress of not knowing if we are asymptomatic/mild symptomatic carriers is unnerving*”; as well as unknowingly spreading it to family members: “*I can’t hug my parents out of fear that they might get sick. Even though I have no or light symptoms I can’t guarantee that I’m not able to spread the virus*”.

Many parents expressed the uncertainty related to COVID-19 as a key stressor. Some parents were stressed due to the uncertain effect of the virus: “*The stress of getting the virus and not knowing how our bodies will react to that. In some cases people have mild symptoms but nobody knows what will happen*”. This uncertainty was also reported from the perspective of the children: “*She also seems somewhat concerned for the health of her grandparents as she’s aware that COVID-19 is more dangerous for older populations*”. Some parents reported being stressed from the uncertainty of how life is changing or will change due to the pandemic: 

“*The uncertainty how COV19 will change social development for our children and how COVID-19 will affect schools and daily life once quarantine and social distancing has past, sometimes can worry us about the impact the virus will have”; while other parents felt uncertain about how COVID-19 will affect their job security: “Uncertainty if husband will return to work in the fall and uncertainty of our child will begin kindergarten in September is very overwhelming*”. 

Job stability as a stressor was also related to financial stability. Several parents reported a loss of income, such as job loss or business closure: “*One of our family members who lives with us and pays rent with us has lost his job, and that instability has made us worry about paying bills*”, and “*[Partner] has had to shut down her restaurant indefinitely which has added a lot of stress to our lives*”. 

#### 3.4.4. Strategies to Cope with Changes

Several parents reported coping strategies they use to manage stress and life changes due to COVID-19 restrictions including being more mindful and focusing on the positive aspects of having more time together as a family: “*We are loving [the] family time and being grateful we all have each other*”; taking prescription medication or seeking therapy: “*I started taking antidepressants to help with anxiety as a result of this*”; and following their pre-COVID-19 routines: “*we have strict routine, not much has changed*”. 

Parents also reported strategies used to increase physical activity, such as scheduling outdoor playtime: “*We try to get outside every day and aim for some active play*”. Some parents identified that getting outside helps with their child’s sleep: “*For sleep, she is not as tired at night but walking at night…helps sleep*”. Lastly, parents also reported coping strategies related to screen time. One parent said they schedule screen time to decrease it. On the quality of screen time, some parents reported that they allowed more screen time so their child can connect with friends and family: *“[Child’s name] spends more time on a device as she can talk with her friends that way*”; and one parent said that screen time, while greater, is shared as a family: “*There is more tv... but always as a family*”. 

#### 3.4.5. Helpful Resources

Specific resources that were frequently mentioned included resources on engaging children in physical activity or any other activities, e.g., crafts, cooking to decrease screen time, tips for grocery shopping during COVID-19, homeschooling, scheduling, and time management. Table 3 includes a summary of resources parents requested to support them through the challenges related to COVID-19.

## 4. Discussion

Our study aimed to understand how health-related behaviors and level of stress, financial and food security have been impacted by the COVID-19 pandemic among a sample of Canadian families with young children. This is one of the first studies in Canada to identify the impact that COVID-19 has had on health-related behaviors and stress levels among families with young children. These findings will help inform how best to support families with young children during this unprecedented time.

One of the most substantive health behavior changes was screen time; 87% of children increased recreational screen use since COVID-19, which is similar to results among studies exploring changes in screen time among older children and adolescents [5]. A systematic review by Stiglic and Viner [24] exploring the health harms from excessive screen use in children and adolescents observed that higher levels of screen time to be strongly associated with greater adiposity, less healthful diets, depressive symptoms as well as lower quality of life. In our sample, while parents expressed concern about the amount of screen time their children were getting, many parents also identified structural changes that made it challenging to limit their children’s screen time such as using screens for children’s online learning/schooling and the need to use screens to engage their children while they completed their paid work from home. Parents in our sample requested ideas of screen-free activities that they could do with their children as ways to reduce their children’s screen time and increase their children’s physical activity. Our results suggest that guidance and resources to help parents manage their children’s screen time and physical activity during this time should acknowledge these structural challenges and should focus on balancing screen time with physically active, screen-free activities versus recommendations to meet screen time and movement guidelines. 

In a recent study of 41 children and adolescents (mean age = 13 years) in Verona, Italy, Pietrobelli and colleagues [6] observed a significant increase in potato chip, red meat, and sugary drink intakes during the COVID-19 lockdown. Although our measures of dietary intake differ from those of Pietrobelli and colleagues [6], our study also found that many families reported eating more food and more snack foods (such as chips or cookies) since the COVID-19 pandemic. Given that many snack foods are high in added sugar, saturated fat, and sodium, this increase in snack food consumption may be of concern. Higher snack food consumption has been found to be associated with increased obesity risk among children and adolescents and increased risk for chronic disease among adults [25]. Although many families reported increases in unhealthful behaviors, there were also notable healthful eating changes being reported since the pandemic. Roughly half of the parents and children whose eating had changed since the pandemic reported eating less fast food/take out. Many families also reported spending more time cooking, making more meals from scratch, eating with their children more often, as well as involving children more often in meal preparation. A survey of 300 adults in the US observed similar trends in cooking more from scratch and dining out less [26]. A 5-year longitudinal study by Larson et al. [27] demonstrated that involving adolescents in meal preparation appears to have a lasting positive influence on diet quality as well as the enjoyment of cooking in young adulthood. These healthful meal preparation habits may result in families nurturing healthy eating and meal preparation habits from a young age. As requested by parents in our sample, resources should be provided to help families with young children continue these positive behavior changes as COVID-19 physical distancing restrictions are lifted, parents return to work outside the home and children return to school. In addition, it will be important to measure how these meal preparation and routines change over time as COVID-19 restrictions are lifted in order to identify which families may need additional supports to sustain children’s engagement in meal preparation. 

A pre-pandemic American survey indicated a national perceived stress average of 4.9 out of 10 [28]. Parents surveyed during a COVID-19 poll reported an increase to 6.7 out of 10 [29], which is similar to the mean stress level found in our sample. Children in this sample seem to be largely protected from fears of COVID-19, as indicated by nearly 50% of respondents saying their child was experiencing “very little” worry and only 7% reporting a high concern. It is possible that other stressors such as changes in routines, social isolation from friends, and adjusting to online schooling may be present, but were not captured in our survey. Some families reported that their children were misbehaving and acting out more often since the pandemic. It could be possible that some children are misbehaving in response to the stress associated with the abrupt shift in their day-to-day routines. In past public health crises, longer durations of quarantine were associated with worsening mental health [30] and so it will be important to assess changes in family-level stress over time. Considering the physiological impact of stress on body function [31] alongside the increased risk of chronic diseases [32], minimizing family stress should continue to be a top priority in COVID-19 response plans and simultaneous efforts made to reduce the chances of such adversities. 

Limited financial or food resources are substantial stressors that many Canadians may face during the COVID-19 pandemic. Many Canadians have experienced reduced work hours or job loss during the pandemic and parents’ ability to work is further complicated by school and child-care center closures. In our sample, approximately 20% of families reported concerns about paying their mortgage, rent, or other bills on time over the next six months. Job stability and financial concerns were also identified in the open-ended responses as key stressors among families. Our study also found that approximately 8% of families reported concerns about having enough money to purchase food for their family over the next six months. This is slightly lower than the national estimate of 12% of families experiencing food insecurity [33]. Our sample consists of a large proportion of families with relatively high household income, which may explain these lower levels of food security concerns. Canadian food banks have experienced a 20% increase in demand, with some areas such as Toronto experiencing a 50% increased demand [34]; a doubling of global food insecurity is predicted by the end of 2020 as a result of COVID-19 [35]. Despite reports of panic-shopping in the early days of the pandemic [3], Canada has a robust food supply system and government officials have assured the public that there is no anticipated food shortage as a result of COVID-19. These public messages may have helped to reassure parents in this sample, leading to a lower concern for food security observed in this study. Given that families with fewer financial and food resources are especially hard-hit in bio-emergencies [36], a focus is needed on the long-term impact of COVID-19 among low-income families and on policies that ensure all Canadian families have adequate financial resources to weather these shocks to our economic system. 

While our study provides an early understanding of the impact that COVID-19 may have on the health and wellbeing of families in Canada, some limitations should be considered when interpreting our results. First, our sample was predominantly Caucasian and roughly 56% had a household income over $100,000, which limits the generalizability of our results to lower-income and racial minority families. Second, all data were self-reported by parents and included parent perception of behavior change (vs. direct measure). The fact that 87% of parents reported an increase in their children’s screen time, which is not typically considered a desirable behavior, may however suggest that our results were not substantively influenced by social desirability bias. Finally, our study did not assess health outcomes, so it is unknown how the stressors and health behaviors assessed in this study are associated with family health status. 

## 5. Conclusions

Among this sample of 254 families with young children, we found that the majority of families reported increased screen time and decreased physical activity since COVID-19 physical distancing restriction implementation. Over half of the families also reported changes to their eating habits and meal routines with some of those changes likely leading to improved diet quality, such as eating more meals from scratch, while others would be expected to lead to lower quality diets, such as eating more snack foods. Parents reported moderately high levels of stress; challenges related to balancing work with the added responsibility of homeschooling, concerns over contracting the COVID-19 virus, employment, and financial instability were identified as key stressors affecting families since the pandemic. Longitudinal research is needed to understand the impact of these behavior changes and stressors on health and weight-related outcomes among families. Our results suggest that COVID-19 response plans should include a focus on ensuring adequate income, minimizing family stress, and supporting healthful eating, activity, and screen-time behaviors among all Canadian families. 

## Figures and Tables

**Figure 1 nutrients-12-02352-f001:**
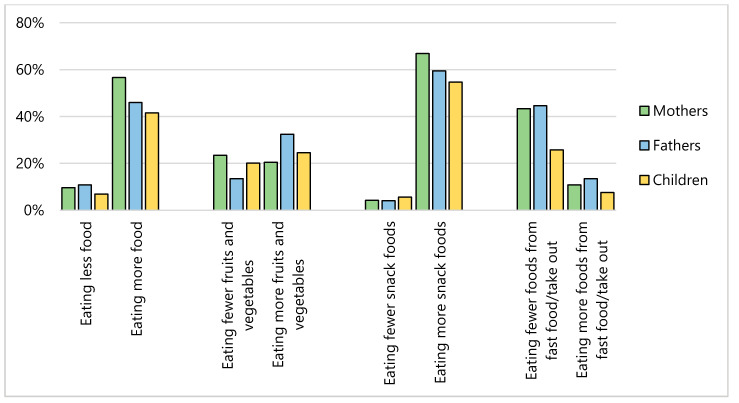
Change in eating behaviors among mothers (*n* = 166), fathers (*n* = 74), and children (*n* = 159) since COVID-19.

**Figure 2 nutrients-12-02352-f002:**
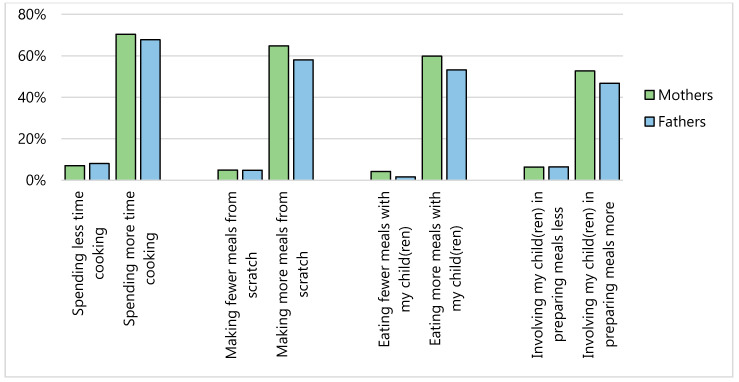
Reported changes in meal routines among mothers (*n* = 142) and fathers (*n* = 62) since COVID-19.

**Figure 3 nutrients-12-02352-f003:**
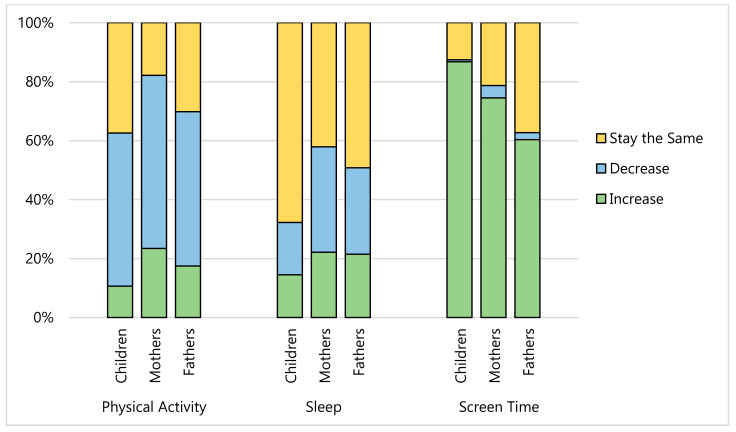
Change in physical activity, sleep and screen time behaviours among mothers (*n* = 235), fathers (*n* = 126), and children (*n* = 310) since COVID-19.

**Table 1 nutrients-12-02352-t001:** Demographics and current health behaviors among families (*n* = 254).

Variables	Household	Children	Mothers	Fathers
	(*n* = 254)	(*n* = 310)	(*n* = 235)	(*n* = 126)
Age, Years, Mean (SD)	-	5.7 (2.0)	37.5 (4.8)	39.4 (5.5)
Ethnicity, *n* (%) ^1^				
Caucasian	-	-	204 (86.8)	111 (88.1)
African American	-	-	2 (0.9)	0 (0.0)
Latin American	-	-	7 (3.0)	3 (2.4)
Asian	-	-	11 (4.7)	5 (4.0)
South/West Asian	-	-	7 (3.0)	4 (3.2)
Other	-	-	3 (1.3)	1 (0.8)
Household Income, *n* (%) ^2^				
Less than $30,000	13 (5.1)	-	-	-
$30,000-$59,999	25 (9.8)	-	-	-
$60,000-$99,999	55 (21.7)	-	-	-
$100,000 or more	144 (56.7)	-	-	-
Marital Status, *n* (%) ^3^				
Single	-	-	7 (3.0)	0 (0.0)
Cohabiting			23 (9.8)	14 (11.1)
Married	-	-	195 (83.0)	104 (82.5)
Separated	-	-	8 (3.4)	1 (0.8)
Physical Activity, Mean (SD) ^4^				
MVPA, hours/week	-	-	4.5 (4.2)	9.5 (13.4)
Time spent walking, hours/week	-	-	4.5 (5.1)	6.1 (8.5)
Time spent sitting, hours/day	-	-	6.1 (2.9)	6.5 (3.1)
Time spent outdoors, hours/day	-	1.1 (0.5)	-	-
Time in active play, hours/day	-	0.9 (0.5)	-	-
Sleep Duration, Mean (SD)				
Hours	-	10.9 (0.7)	8.2 (1.1)	7.9 (0.9)
Screen Time, Mean (SD)				
Hours/day	-	2.4 (1.6)	2.7 (1.6)	2.8 (1.7)
Eating Patterns, Mean (SD)				
Fruit and Vegetable, times/day	-	4.5 (2.2)	4.1 (2.1)	3.8 (2.2)
Snack Foods, times/day	-	0.8 (0.8)	1.0 (0.9)	0.9 (0.9)
Fast Food/Take-Out, times/week	-	-	0.8 (0.8)	0.8 (0.7)

^1^ 3 parent respondents did not report their ethnicity. ^2^ 17 parent respondents did not disclose their household income. ^3^ 9 parent respondents did not report marital status. ^4^ Sample size for parent physical activity varies due to non-response of quantity or units.

**Table 2 nutrients-12-02352-t002:** Reported stress, financial, and food security among families (*n* = 254).

Variables	Children (*n* = 310)	Mothers (*n* = 235)	Fathers (*n* = 126)
Stress Level 1-10, Mean (SD)	-	6.8 (1.9)	6.0 (2.5)
Child Concern COVID-19, *n* (%) ^1^			
Very little	153 (49.4)	-	-
Somewhat	117 (37.7)	-	-
Very Much	22 (7.1)	-	-
Financial Stress, *n* Yes (%)			
During the past month	-	45 (19.1)	17 (13.5)
Over the next 6 months	-	51 (21.7)	22 (17.5)
Food Insecurity, *n* Yes (%)			
During the past month	-	20 (8.5)	6 (4.8)
Over the next 6 months	-	20 (8.5)	6 (4.8)

^1^ Some parent respondents (n = 18) selected ‘I don’t know’ or ‘I prefer not to answer’.

**Table 3 nutrients-12-02352-t003:** Selected quotes illustrating resources parents perceived as helpful.

Resources on	Quotation
Increasing physical activity	*“* *More exercise activities for inside…that last for around 1/2 hr.”* *“Resources around fun physical activity they can do indoors, or challenges they can participate in”*
Reducing screen time	*“* *Ideas to get the kids off screens”* *“Any tips for keeping a 4-year-old independently busy (and not on a screen) while mommy gets work done would be great too!!”*
Homeschooling	*“Having a better way to ease [daughter] into her homework would have helped at first to minimize the appearance of a school-related pressure one encounters as a parent at first.”*
Time management	*“I’m trying to figure out the balancing helping with school work and working myself fulltime.”* *“More ready-to-go day plans for pre-school aged children.”*
Parenting	*“* *We could use some support around managing tantrums and fighting between siblings.”*
Grocery shopping	*“How to plan ahead and minimize the trips to the grocery store?”* *“Ideally I’d like to only go every 2 weeks, but don’t know how to stretch out fresh produce to last that long.”*
Cooking and diet	*“* *Meal planning tools and trackers for the # of servings of healthy foods we eat each day vs. the # of unhealthy snacks or “treats””* *“Activity ideas to help get (child’s name) interested in helping in the kitchen”*
Understanding COVID-19	*“How to talk to kids about not being able to do things they look forward to ie. seeing grandparents, having a birthday party, playing with friends.”* *“Kids directed video about Corona virus, I didn’t realize he was so worried until I asked him while doing the questionnaire. I talk with him about it often, in a non-scary way.”*
Improving mental health	*“Mindfulness activities or other stress-reduction methods, appropriate for all ages.”* *“Ideas for emotional regulation”* *“We need supports on how to support the kid’s anxiety during this time.”*

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
