# Peer review of "The Impact of COVID-19 on Health Behavior, Stress, Financial and Food Security among Middle to High Income Canadian Families with Young Children"

_nutrients, 2020, doi:10.3390/nu12082352_

Round 1

Reviewer 1 Report

The manuscript, The impact of COVID-19 on health behavior, stress, financial and food security among Canadian families with young children is timely and should be of interest to the readership. It is clear that this is a unique sample (predominantly Caucasian with over half of the sample with income over $100,000) as stated in lines 387-388. This should be stated in the title (middle to high income Canadians families with young children). This is an important group to study as quite often the focus in on low income populations. 

In the study design it was stated that to be eligible to participate, families must have one child between 18 months and 5 years of age. The results stated the average age for children was 6 years with a standard deviation of 2.0 years. Some clarification is needed here. 

Much of the result section is repeated in the first paragraph of the discussion (lines 304 - 320). The discussion of screen time is somewhat superficial. Inclusion of the consequences of this increased screen time on health behaviors should also be included. Similarly, what are the potential health consequences on the increase consumption of snack foods during this period. 

The results of this study showed that children were protected from the stress of COVID-19. It seems that parents in this sample are absorbing most of this stress. What are some of the consequences of the increased family stress mentioned in lines 243-279? Exploring this area may provide indication/information on the type of intervention needed.  

Sections of the discussion (mentioned above) could be further developed. 

A very timely manuscript with very relevant information. 

Author Response

Reviewer #1:

The manuscript, The impact of COVID-19 on health behavior, stress, financial and food security among Canadian families with young children is timely and should be of interest to the readership. It is clear that this is a unique sample (predominantly Caucasian with over half of the sample with income over $100,000) as stated in lines 387-388. This should be stated in the title (middle to high income Canadians families with young children). This is an important group to study as quite often the focus in on low income populations.

Response: Thank you for this kind feedback and for your constructive review of our manuscript. We have edited our title to include that our sample is predominantly middle to high income families. This change has been highlighted on lines 3-4.

In the study design it was stated that to be eligible to participate, families must have one child between 18 months and 5 years of age. The results stated the average age for children was 6 years with a standard deviation of 2.0 years. Some clarification is needed here.

 Response: Thank you for bringing this to our attention. To clarify, families were eligible if they had a child between 18 months and 5 years of age at the time of registration for the study. Since this is a longitudinal study and our sample includes families from two pilot phases and the full study (as mentioned in lines 80-82), the child participants would be older as highlighted in our results (re: average age of 6 years). Line 78 has been edited to include:

“… they had at least one child between 18 months and 5 years of age at the time of registration for the study…”

Much of the result section is repeated in the first paragraph of the discussion (lines 304 - 320). The discussion of screen time is somewhat superficial. Inclusion of the consequences of this increased screen time on health behaviors should also be included. Similarly, what are the potential health consequences on the increase consumption of snack foods during this period.

Response: We have removed most of the repeated information from the opening paragraph in the discussion. The first paragraph (lines 326-331) instead includes our study aims as well as the precedence of this research. We agree that including the consequences of increased screen time is needed, especially as the child participants on average were receiving more than double above the recommendations by CSEP. We have made these additions to lines 334-337:

“A systematic review by Stiglic and Viner [24] exploring the health harms from excessive screen use in children and adolescents observed that higher levels of screen time to be strongly associated with greater adiposity, less healthful diets, depressive symptoms as well as lower quality of life.”

In addition, we have added the potential health consequences of increased snack food consumption to lines 381-384:

Given that many snack foods are high in added sugar, saturated fat and sodium, this increase in snack food consumption may be of concern. Higher snack food consumption has been found to be associated with increased obesity risk among children and adolescents and increased risk for chronic disease among adults [25].

The results of this study showed that children were protected from the stress of COVID-19. It seems that parents in this sample are absorbing most of this stress. What are some of the consequences of the increased family stress mentioned in lines 243-279? Exploring this area may provide indication/information on the type of intervention needed. 

Response: We have further described some of the consequences of increased stress (line 410-412):

“Considering the physiological impact of stress on body function [31] alongside the increased risk of chronic diseases [32] ...”

 In addition, our closing sentence in this paragraph now includes that (lines 412-413):

“…minimizing family stress should continue to be a top priority in COVID-19 response plans as the simultaneous efforts to reduce the chances of such adversities.”

Sections of the discussion (mentioned above) could be further developed. A very timely manuscript with very relevant information.

Response: Thank you again for indicating which areas of the discussion need to be further developed. All changes to the discussion are highlighted in the updated manuscript.

(x) English language and style are fine/minor spell check required

Response: Thank you for highlighting this in the Review Report Form. We have done a thorough review of the language and style. Changes have been highlighted with the Track Change feature.

Reviewer 2 Report

This was a good paper to read. There were a few grammatical errors, which I suggest the authors go through with a fine-toothed comb the article to catch such issues. 

Per the authors, there are issues with generalizability. Not much you can do, though, ex post facto. 

The only major thing I would consider to change or edit is the future directions as it pertains to health outcomes. You should discuss throughout the paper that these findings are upstream and we need to pivot to now be aware of downstream events as a result of changed behavior. This would be of utmost importance. 

Author Response

Reviewer #2:

This was a good paper to read. There were a few grammatical errors, which I suggest the authors go through with a fine-toothed comb the article to catch such issues.

 Per the authors, there are issues with generalizability. Not much you can do, though, ex post facto.

Response: Thank you for your constructive review of our paper. We have done a thorough review of the language and grammar. Changes have been highlighted with the Track Change feature. In addition, we mention this limitation in the last paragraph of our discussion section. Reviewer #1 also mentioned that our title should further describe that this sample was predominantly middle to high income Canadian families. We have also highlighted in our discussion that our results may not be generalizable to lower income families.

The only major thing I would consider to change or edit is the future directions as it pertains to health outcomes. You should discuss throughout the paper that these findings are upstream and we need to pivot to now be aware of downstream events as a result of changed behavior. This would be of utmost importance.                                       

Response: We agree that making this distinction is very important. Reviewer #1 recommended including the consequences of increased screen time, snack food consumption, and stress among parents would build for a stronger discussion and we have added text addressing this recommendation. In addition to your point above, we have added the following to line 413 to note such downstream events:

“… as the simultaneous efforts to reduce the chances of such adversities.”

(x) English language and style are fine/minor spell check required

Response: Thank you for highlighting this in the Review Report Form. As mentioned above, we have done a thorough review of the language and style.

Does the introduction provide sufficient background and include all relevant references? (x) Can be improved

Response: Thank you for mentioning this in the Review Report Form. We have reviewed our introduction to ensure we provide sufficient background and include all relevant references.
